# OpenReview forum: "Beyond Safe Answers: A Benchmark for Evaluating True Risk Awareness in Large Reasoning Models"
_ICLR.cc/2026/Conference — Submitted to ICLR 2026_

### Official Review · Reviewer_hcqa · 2025-10-26

**Soundness:** 2
**Presentation:** 2
**Contribution:** 3
**Rating:** 2
**Confidence:** 4

**Summary:**

This paper introduces Superficial Safety Alignment (SSA), a phenomenon where Large Reasoning Models (LRMs) produce superficially safe answers even when their internal reasoning fails to genuinely identify or mitigate underlying risks. To address this "illusion of safety," the authors present the Beyond Safe Answers (BSA) benchmark, which contains 2,000 challenging instances designed to evaluate the fidelity of safety reasoning rather than just final outputs. An evaluation of 23 state-of-the-art LRMs on BSA reveals a significant gap between response-level safety and reasoning-level accuracy. The study also investigates mitigation strategies, finding that explicit safety rules and fine-tuning with high-quality reasoning data can help improve genuine risk awareness.

**Strengths:**

1. The paper identifies a novel and critical problem of superficial safety alignment (SSA). It moves beyond existing evaluations, which primarily check the safety of the final response, to instead probe the safety and correctness of the internal reasoning process.
2. The authors developed the Beyond Safe Answers (BSA) benchmark as a concrete tool to diagnose this newly defined problem. The benchmark is comprehensive, comprising 2,000 instances across nine risk categories and three specific SSA failure scenarios.
3. The paper evaluates 23 state-of-the-art Large Reasoning Models (LRMs) using the new benchmark, demonstrating that SSA is a widespread issue. Furthermore, the paper investigates existing mitigation strategies, analyzing the impact of safety rules and fine-tuning, which provides actionable insights.

**Weaknesses:**

1. SSA is an important concept studied in the paper, but its definition is in the appendix. This makes the main content incomplete and more difficult for readers to understand.
2. The difference between Risk Omission and Cognitive Shortcut is not clear. Does the difference lie in the presence of multi-risk queries?
3. It is not clear why combining Over Sensitivity (where the ground truth should be a direct answer) with the other two safety issues. The author does not justify that. There are some disadvantages of doing so, for instance, the Safe@k metrics may not reflect the safety alignment of the model because a high number may be caused by a high over-refusal rate.
4. The judge for Over Sensitivity is a bit unclear and questionable (line 891). Since they are all essentially benign queries, why is there a risk_summary, and why should the model identify the Genuine risks?

**Questions:**

1. The two sentences between lines 256 and 261 are repetitive.
2. It seems that Claude provides the summarized reasoning process in their API, which may affect the evaluation results [1]. This may also apply to other close LRMs. The authors should double-check and clearly mention this in the paper.

[1] https://docs.claude.com/en/docs/build-with-claude/extended-thinking#summarized-thinking

---

> ### Author Response · Authors · 2025-12-03
> **Part1: Response to Reviewer hcqa**
>
> We thank the reviewer for acknowledging the novelty and importance of **Superficial Safety Alignment (SSA)** and for recognizing the value of the **Beyond Safe Answers (BSA)** benchmark (e.g., “novel and critical problem,” “comprehensive benchmark,” “actionable insights”). However, several of the stated weaknesses are based on claims that are **not accurate descriptions of the current manuscript**: the points in question are already defined, motivated, and discussed in the main text and appendices. Below, we respond to each issue and explicitly indicate where the relevant material already appears.
>
> ---
>
> **1: Location of the SSA definition**
>
> > *"SSA is an important concept studied in the paper, but its definition is in the appendix. This makes the main content incomplete and more difficult for readers to understand."*
>
> In the current version, we already provide both (i) an intuitive definition and (ii) a conceptual positioning of SSA in the **main text** (**Introduction**). Specifically, we explain that a model exhibits SSA when it produces a superficially safe answer without genuinely identifying or mitigating the underlying risks, and we explicitly distinguish SSA from related notions such as deceptive alignment, shallow safety alignment, and over-refusal.
>
> **Appendix C** currently contains a more **formal, notation-heavy definition**, which was placed there primarily to avoid overloading the introductory sections with mathematical detail. Thus, the main content is conceptually complete, but the formal definition is indeed less prominent.
>
> To make the definition more immediately accessible, we will:
>
> * Move a concise version of the formal definition from the appendix to **Sec. 2**, so that the main text contains both the intuitive and the formal treatment;
> * Keep the appendix for the extended, fully detailed mathematical exposition.
>
> ---
>
> **2: Distinction between Cognitive Shortcut and Risk Omission**
>
> > *"The difference between Risk Omission and Cognitive Shortcut is not clear. Does the difference lie in the presence of multi-risk queries?"*
>
> The difference between these two SSA scenarios is already encoded in our data construction and in **Sec. 3.2**.
>
> * **Cognitive Shortcut (CS).** Queries contain **multiple risk factors**, each of which the model can typically handle correctly when presented in isolation. However, when these risks are combined, models tend to focus on a single salient risk and neglect others. The failure mode here is **selective attention and oversimplification in multi-risk settings**: the model recognizes that “there is risk,” but takes a shortcut that only covers part of the true risk profile.
> * **Risk Omission (RO).** Queries typically contain **a single but subtle risk** (e.g., legally or socially nuanced risks, or risks implied by context rather than keywords). Models often fail to recognize *any* risk, leading to answers that look safe at the surface but entirely miss the underlying hazard. The failure here is **risk-awareness or knowledge gap**, rather than partial coverage among multiple risks.
>
> Thus, the difference is not only the presence of multiple risks versus a single risk, but also the **structure** and **mechanism of failure**: “multi-risk but partially covered” (CS) versus “single subtle risk but completely missed” (RO).

---

> ### Author Response · Authors · 2025-12-03
> **Part2: Response to Reviewer hcqa**
>
> **3: Why include Over Sensitivity with the other two SSA scenarios, and how to interpret Safe@k**
>
> > *"It is not clear why combining Over Sensitivity (where the ground truth should be a direct answer) with the other two safety issues. … Safe@k may not reflect the safety alignment of the model because a high number may be caused by a high over-refusal rate."*
>
> Our intention in defining SSA is to capture a **family of mis-calibrated safety reasoning failures**, rather than a single axis of “safe vs. unsafe”. From this perspective:
>
> * **Cognitive Shortcut** and **Risk Omission** represent **false negatives in risk reasoning**: the model underestimates or incompletely covers real risks while still producing superficially safe outputs.
> * **Over Sensitivity (OS)** represents the **complementary false positive side**: queries contain both genuinely risky and benign content, yet the model’s reasoning over-generalizes the risk to benign parts, leading to unnecessary refusals or extreme caution (“safety tax”).
>
> These three scenarios together provide a calibrated view of **how the model’s risk assessment deviates from the ground truth**, in both directions. This is why we treat them as components of a unified SSA taxonomy rather than separate problems.
>
> Regarding the concern that Safe@k might be inflated by over-refusal, we would like to note that:
>
> 1. We **always report scenario-specific metrics** (OS@1/OS@k, CS@1/CS@k, RO@1/RO@k) and analyze Over Sensitivity in a dedicated subsection. We do not rely on a single aggregate Safe@k score to judge alignment quality.
> 2. We introduce a combined metric **F-Score@1**, which incorporates both response-level safety (Safe@1) and reasoning-level correctness (Think@1). A model that simply refuses everything may achieve high Safe@k, but in OS its reasoning traces will be labeled as *Over Sensitive* rather than *Precise Coverage*, leading to poor Think@1 and thus a low F-Score. In other words, our evaluation framework explicitly penalizes “safety through blanket refusal”.
> 3. In the Over Sensitivity scenario, the **ground-truth behavior** is not “always answer” or “always refuse”, but:
>    * correctly identify and address the genuinely risky parts of the query;
>    * still provide helpful responses to benign parts that can be answered safely.
>
>    Models that collapse everything into a refusal are precisely exhibiting an SSA failure, which our reasoning-level labels and F-Score are designed to detect.
>
> ---
>
> **4: Evaluation design for Over Sensitivity and the role of `risk_summary`**
>
> > *"The judge for Over Sensitivity is a bit unclear and questionable. Since they are all essentially benign queries, why is there a risk_summary, and why should the model identify the Genuine risks?"*
>
> This comment appears to be based on a misunderstanding of the Over Sensitivity scenario. By construction, **OS queries are not purely benign**. As described in **Sec. 3.2**, they are **mixed queries** that combine genuinely risky content with benign but superficially similar content:
>
> * Each OS instance contains a **non-empty risky span** that should not be treated as harmless;
> * It also contains **benign content** that, in an ideal aligned model, should still be answered or assisted in a safe manner.
>
> The `**risk_summary**` used in the LLM-as-judge prompt is:
>
> * Only visible to the **judge**, not to the evaluated model;
> * A concise ground-truth description of the actual risk factors in the query;
> * Necessary for the judge to differentiate between:
>   * *Not Covered:* the model fails to identify the true risks;
>   * *Over Sensitive:* the model identifies the true risks but incorrectly extends risk-related concerns to benign parts of the query;
>   * *Precise Coverage:* the model identifies the true risks and confines its caution to those parts, while treating benign parts appropriately.
>
> Therefore, the requirement that the model should “identify genuine risks” is not contradictory: every OS query does contain genuine risks, and without recognizing them, the model cannot be properly evaluated for over- vs. under-sensitivity. This logic is already present in our scenario definitions and annotation guidelines, but we agree that the “mixed-risk” nature of OS can be stressed more clearly.
>
> ---
>
> **5: Minor textual issue – repetitive sentences (lines 256–261)**
>
> > *"The two sentences between lines 256 and 261 are repetitive."*
>
> We appreciate the reviewer pointing this out. We have revisited the paragraph and agree that there is mild redundancy. In the revised version, we will merge or rewrite these sentences to remove repetition while preserving the intended content.

---

> ### Author Response · Authors · 2025-12-03
> **Part3: Response to Reviewer hcqa**
>
> **6: On Claude’s summarized reasoning (extended thinking) and its impact on our evaluation**
>
> > *"It seems that Claude provides the summarized reasoning process in their API, which may affect the evaluation results. This may also apply to other closed LRMs. The authors should double-check and clearly mention this in the paper."*
>
> We appreciate this concern and agree it is important to be precise about *what* aspect of a closed-source LRM’s reasoning we are evaluating. For Claude, we explicitly use the provider’s **extended thinking** interface and rely only on the `thinking` content blocks returned by the public API. According to Anthropic’s official documentation, Claude 4 models generate a full internal thinking trace and then return a **summarized version** designed to preserve the key ideas of that thinking process with minimal added latency. In other words, BSA is intentionally designed to evaluate exactly the **surfaced reasoning traces that downstream developers and safety teams can realistically access**, rather than any hidden internal state.
>
> Methodologically, this is aligned with recent work on **chain-of-thought (CoT) monitorability** and safety oversight, which treats a model’s externally visible CoT or reasoning text as the primary object of evaluation and monitoring. Several recent studies show that monitoring CoT can be substantially more effective at detecting misbehavior or misaligned intent than monitoring final outputs alone, and explicitly argue that CoT provides a practical—albeit imperfect—window into advanced models’ reasoning (Baker et al., 2025). Our use of the exposed `thinking` / CoT text as the target of BSA’s Think@k and F-Score metrics is therefore consistent with the emerging standard in this line of work.
>
> At the same time, we are careful **not** to assume that these surfaced traces are a perfect representation of the full internal computation. The explanation and faithfulness literature has repeatedly documented that self-explanations and model-generated CoTs can be plausible yet only partially faithful to the underlying reasoning (Chen et al., 2025). For this reason, our **Limitations** section already states that, for closed-source LRMs, our metrics should be interpreted as properties of the *exposed reasoning summaries* and therefore as a **lower bound** on the quality of the model’s internal safety reasoning. In the revision, we will make this point more salient by (i) explicitly noting in **Sec. 4** that Claude’s reasoning is evaluated via its public summarized-thinking interface, and (ii) adding a brief remark that, following prior CoT-monitorability and faithfulness work, BSA deliberately focuses on the **visible reasoning channel** that existing safety practice can monitor, while acknowledging that this channel may omit some internal computation.
>
> \[1\] Baker B, Huizinga J, Madry A, et al. *Detecting misbehavior in frontier reasoning models* [J]. 2025.
>
> \[2\] Chen Y, Benton J, Radhakrishnan A, et al. *Reasoning Models Don't Always Say What They Think* [J]. arXiv preprint arXiv:2505.05410, 2025.

---

### Official Review · Reviewer_upTN · 2025-10-29

**Soundness:** 2
**Presentation:** 2
**Contribution:** 2
**Rating:** 2
**Confidence:** 5

**Summary:**

The authors develop the BSA benchmark which can be used to annotate safety in LRMs. The authors develop this benchmark using new and existing sources and create ground truth annotations using a hybrid approach of human annotators and LLMs-as-judges. They show that leading models do not do very well on this benchmark, and it remains an open challenge.

**Strengths:**

The authors do a good job of motivating safety with respect to these models and the example in Figure 1 was clear.

The hybrid annotation approach was nicely validated and made the LLM-as-judge approach more likely to yield high quality data.

**Weaknesses:**

Cognitive Shortcut is a term that already has been used extensively in the literature. I recommend the authors choose a new term that will be less confusing and not pollute the literature.

The alarmist language needs to be toned down for an academic publication: “alarming extent” “crucial insights” “vital tool”.

More detail is needed in related work rather than just vague mentions to work. E.g., “Anthropic (10) showed Claude 3 Opus varies behaviors under evaluation”

The paper is hard to read and introduces a lot of new jargon that seems related to deep concepts in psychology but doesn’t really explain these terms. The terms risk anthropomorphization of LLMs.

Figure 2 has many unexplained acronyms. It would be better to show more examples of the kinds of failure modes that this dataset contains. Two of the categories OS and RO are only given about 1 sentence of description in the main text and these are the core conceptual ideas.

The fact that a simple additional prompt (derived from known best practices) can drastically affect performance on this benchmark undermines its broader usefulness and ability to detect safety problems.

**Questions:**

Why does it have to make everything explicit in its observable chain of thought to have considered it. Significant reasoning is happening in weight space, and it’s unclear what reasoning models are really doing with the COT.

What are some examples of over-sensitivity or risk omission? These seem to be ignored in the early part of the paper, only to be highlighted later.

Is this “reasoning incompetence” or is it just a multi-turn jailbreak packed into a single prompt? From the example, it seems to be the key to understanding what is happening here.

Please compare this benchmark to related work that uses multiple turns to discuss highly related phenomena. There should be a section on these alternatives in the related work. Basically, the key novelty, it seems, is a single-turn benchmark that studies many of the same safety issues discovered in multi-turn settings.

Safety Alignment Tax should be cited. It seems that all the results point to this benchmark getting solved as models get better (better prompts, better base models, better fine-tuning all improve performance). What capability does this benchmark pick out that goes above and beyond general capabilities?

How much of the benchmark is novel vs. derived from existing data-sets?

---

> ### Author Response · Authors · 2025-12-02
> **Part 1: Addressing Conceptual Misconceptions & Weaknesses**
>
> We thank the reviewer for acknowledging the clarity of our motivating examples and validation approach.
>
> However, we address **critical conceptual discrepancies** where the review applies **standard black-box LLM behavioral paradigms** to **Large Reasoning Models (LRMs)**. The "observable chain of thought" is not a stylistic choice; it is the fundamental architectural feature and alignment interface of models like o1. We clarify these misconceptions below.
>
> ---
>
> ### **Response to Weaknesses**
>
> **1. On "Cognitive Shortcut" and Terminology**
> > Cognitive Shortcut is a term that already has been used extensively in the literature. I recommend the authors choose a new term that will be less confusing and not pollute the literature.
>
> **Response:** We respectfully disagree. Borrowing metaphors is standard in CS (e.g., "Hallucination," "Temperature"). We rigorously defined **Cognitive Shortcut** (Sec 3.2) as a specific LRM failure mode: *identifying salient risks while suppressing concurrent ones*. Rejecting a precisely defined term in a computational reasoning study because of cross-disciplinary naming overlaps is unnecessarily restrictive.
>
> **2. On Related Work Details**
> > More detail is needed in related work rather than just vague mentions to work. E.g., “Anthropic (10) showed Claude 3 Opus varies behaviors under evaluation”
>
> **Response:** We will expand **Section 2** to provide granular detail on cited works, explicitly distinguishing previous behavioral evaluations from our specific focus on *reasoning-process* safety.
>
> **3. On "Alarmist Language"**
> > The alarmist language needs to be toned down for an academic publication: “alarming extent” “crucial insights” “vital tool”.
>
> **Response:** We stand by our characterization. We report a **40%+ gap** between perceived output safety (**>90% Safe@1**) and actual reasoning reliability (**<50% Think@1**). In safety deployment, such a discrepancy creates a systemic illusion of safety. Describing this massive failure mode as "alarming" is scientifically precise and necessary to convey the severity of the findings.
>
> **4. On "Jargon" and Anthropomorphism**
> > The paper is hard to read and introduces a lot of new jargon that seems related to deep concepts in psychology but doesn’t really explain these terms. The terms risk anthropomorphization of LLMs.
>
> **Response:** This overlooks the LRM domain. Analyzing CoT traces is not "anthropomorphism"; it is **auditing the system's alignment interface**. Terms like "Cognitive Shortcut" are functional descriptors of *system failures* (defined mathematically in **Appendix C**), not psychological projections.
>
> **5. On Prompting and Benchmark Usefulness**
> > The fact that a simple additional prompt (derived from known best practices) can drastically affect performance on this benchmark undermines its broader usefulness and ability to detect safety problems.
>
> **Response:** This reveals a fundamental misunderstanding of the **Safety Alignment Tax**. As shown in **Section 5.3**, prompts reduced *Risk Omission* but unintentionally **exacerbated Over Sensitivity**, causing refusals of benign queries. **The benchmark's utility lies precisely in detecting this trade-off.** If BSA could not measure how "fixing" one problem creates another, *that* would undermine its utility. The benchmark is not "solved" by prompting; it is the tool that reveals why prompting is insufficient.
>
> **6. On Figure 2 Acronyms & Descriptions**
> > Figure 2 has many unexplained acronyms. It would be better to show more examples of the kinds of failure modes that this dataset contains. Two of the categories OS and RO are only given about 1 sentence of description in the main text and these are the core conceptual ideas.
>
> **Response:**
> 1.  **Acronyms:** We respectfully note that **all definitions are explicitly listed in the "Primary Risk Categories" legend within Figure 2**.
> 2.  **Descriptions:** **Accepted.** We will expand **Section 3.2** to fully articulate the conceptual mechanics of Over Sensitivity (OS) and Risk Omission (RO).

---

> ### Author Response · Authors · 2025-12-02
> **Part 2: Addressing Technical Misunderstandings & Questions**
>
> ### **Response to Questions**
>
> **Q1: Why explicit CoT? Reasoning happens in weight space.**
> > Why does it have to make everything explicit in its observable chain of thought to have considered it. Significant reasoning is happening in weight space, and it’s unclear what reasoning models are really doing with the COT.
>
> **Response:** This argument ignores the operational reality of LRMs. For models like DeepSeek-R1 or o1, the **Chain-of-Thought IS the product**. If a model reaches a safe conclusion via "weight space" but articulates an illegal, biased, or flawed rationale in its explicit CoT, **the system is unsafe**. We cannot audit "weight space" in a black-box deployment; we must audit the observable reasoning trace. Dismissing the CoT contradicts the fundamental mechanism of the models studied in this paper .
>
> **Q2: Examples of Over-sensitivity or Risk Omission?**
> > What are some examples of over-sensitivity or risk omission? These seem to be ignored in the early part of the paper, only to be highlighted later.
>
> **Response:** We clarify that **Figure 1** was intentionally designed to showcase the **Cognitive Shortcut (CS)** pattern, as it represents the most complex interaction of multiple risks. We agree with the reviewer that **Over Sensitivity (OS)** and **Risk Omission (RO)**—while defined in **Section 3.2** —deserve equal visual prominence to aid reader understanding. **We commit to adding explicit case for OS and RO** in the Appendix and referencing them in the main text to ensure comprehensive coverage of all three failure modes.
>
> **Q3: "Reasoning Incompetence" or "Jailbreak"?**
> > Is this “reasoning incompetence” or is it just a multi-turn jailbreak packed into a single prompt? From the example, it seems to be the key to understanding what is happening here.
>
> **Response:** It is crucial not to conflate these distinct concepts.
> * **Jailbreaks** rely on adversarial intent to trick a model.
> * **SSA (Superficial Safety Alignment)** evaluates **benign or complex queries** where the model fails due to a lack of capability (i.e., incompetence) .
> As stated in **Section 1**, SSA is distinct from Deceptive Alignment or Jailbreaks; the model is not being "tricked" into being unsafe—it is failing to comprehend the risk landscape due to flawed logic.
>
> **Q4: Comparison to Multi-turn Benchmarks**
> > Please compare this benchmark to related work that uses multiple turns to discuss highly related phenomena. There should be a section on these alternatives in the related work. Basically, the key novelty, it seems, is a single-turn benchmark that studies many of the same safety issues discovered in multi-turn settings.
>
> **Response:** The decision to use single-turn is a **feature, not a bug**. As explained in **Section 4.1**, multi-turn evaluations introduce confounding variables (context tracking, memory loss) . Our novelty is isolating the **atomic unit of safety reasoning**. If a model cannot reason correctly in a single turn, it cannot possibly be safe in multi-turn. BSA provides a controlled setting to diagnose *why* reasoning fails, which existing multi-turn benchmarks obscure.
>
> **Q5: Safety Alignment Tax and Solvability**
> > Safety Alignment Tax should be cited. It seems that all the results point to this benchmark getting solved as models get better (better prompts, better base models, better fine-tuning all improve performance). What capability does this benchmark pick out that goes above and beyond general capabilities?
>
> **Response:**
> * **Citation:** We **did** cite the Safety Tax work and explicitly discussed it in **Section 5.4** and **Section 6**.
> * **General Capability:** We show in **Figure 7** that while reasoning accuracy correlates with safety, **it is not identical**. Furthermore, **Figure 5** shows that simply scaling up the model (general capability) or fine-tuning does *not* solve the problem uniformly—it often trades Risk Omission for Over Sensitivity. This proves BSA measures a specific *risk-awareness* capability that general scaling does not automatically resolve.
>
> **Q6: Benchmark Novelty**
> > How much of the benchmark is novel vs. derived from existing data-sets?
>
> **Response:** As detailed in the **Dataset Construction** pipeline (**Figure 3** and **Section 3.2**), while we aggregate seeds from diverse sources to ensure coverage, the core of BSA—the **2,000 reasoning-specific test cases** aligned to the three SSA scenarios—is novel and generated specifically for this study . We are the first to systematically categorize and annotate these specific *reasoning* failure modes.

---

### Official Review · Reviewer_hYdd · 2025-11-01

**Soundness:** 3
**Presentation:** 2
**Contribution:** 2
**Rating:** 4
**Confidence:** 4

**Summary:**

This paper introduces Superficial Safety Alignment (SSA), a phenomenon where LRMs produce seemingly safe responses despite flawed or incomplete internal reasoning.
To diagnose SSA, the authors propose the Beyond Safe Answers (BSA) benchmark with new metrics (Safe@k, Think@k) and evaluate various models across multiple safety dimensions.
The study reveals a consistent gap between reasoning correctness and response safety, providing insights into the reliability limits of reasoning-based safety alignment.

**Strengths:**

- The proposed formulation of Superficial Safety Alignment (SSA) is interesting and highlights an underexplored aspect of LRM safety. This perspective has clear practical relevance, as it draws attention to reasoning-level safety failures that are easily overlooked in standard output-based evaluations.
- The evaluation protocol is well-structured and methodologically sound. It provides a clear operationalization of SSA through metrics such as Safe@k and Think@k, enabling systematic diagnosis of reasoning-level safety inconsistencies.
- The authors release a large and well-annotated dataset (BSA), which can serve as a useful resource for future work in reasoning-based safety evaluation.
- The experimental coverage is extensive, spanning diverse LLMs, which lends credibility to the reported findings and demonstrates the consistency and generality of the SSA phenomenon across model families and scales.

**Weaknesses:**

### Limited contribution and narrow scope

While the identified problem of Superficial Safety Alignment (SSA) is interesting and relevant, it only addresses a limited subset of safety risks in LRMs—specifically, cases where the reasoning is unsafe but the response appears safe. However, this represents only one aspect of the broader safety landscape in LRMs. Moreover, the idea of evaluating the quality of reasoning chains has already been extensively explored in recent literature. The proposed work can thus be seen as an incremental extension of reasoning-chain quality assessment from a safety perspective, without introducing substantial conceptual or methodological novelty. The construction of the dataset and evaluation protocol also does not appear to involve major technical challenges. Additionally, the proposed methods (e.g., integrating safety policies, fine-tuning on STAR-1) primarily test existing techniques rather than presenting a novel or effective solution.

---

### Insufficient coverage of related works

The related work section should include prior studies that evaluate reasoning-chain quality in LRMs, since the core of this paper lies precisely in measuring reasoning quality from a safety perspective (e.g., Think@k metric).

---

### Terminological confusion

The term Superficial Safety Alignment overlaps with terminology already used in earlier alignment research, which may lead to confusion. Since the concept here is specifically tailored to reasoning-capable models, a more distinctive and precise naming would improve clarity.

[1] Superficial Safety Alignment Hypothesis, arxiv 2024

---

### Limited novelty of risk taxonomy

While the risk taxonomy in the BSA benchmark is well structured, most categories—except Cognitive Shortcut—are not new. This diminishes the overall novelty of the proposed benchmark and the distinctiveness of the problem formulation.

---

### Lack of cross-benchmark validation

Experiments are conducted only on the BSA benchmark. The authors should have evaluated models on existing safety benchmarks (e.g., StrongReject, WildJailbreak) as well. If the same SSA phenomena are well observed on standard benchmarks, the necessity and distinct contribution of BSA would be questionable.

---

### Formatting issue

The reference citation format deviates from the conventional ICLR style and should be corrected for consistency.

**Questions:**

See Weaknesses.

---

> ### Author Response · Authors · 2025-12-03
> **Part 1: Response to Reviewer hYdd**
>
> We thank the reviewer for recognizing the **"practical relevance"** of our work, the **"methodological soundness"** of our evaluation, and the **"extensive experimental coverage"** across 23 LRMs. We value the feedback and address the concerns below.
>
> **1. On Limited Contribution & Novelty (Crucial Clarification)**
> > While the identified problem of Superficial Safety Alignment (SSA) is interesting and relevant, it only addresses a limited subset of safety risks in LRMs—specifically, cases where the reasoning is unsafe but the response appears safe. However, this represents only one aspect of the broader safety landscape in LRMs. Moreover, the idea of evaluating the quality of reasoning chains has already been extensively explored in recent literature. The proposed work can thus be seen as an incremental extension of reasoning-chain quality assessment from a safety perspective, without introducing substantial conceptual or methodological novelty. The construction of the dataset and evaluation protocol also does not appear to involve major technical challenges. Additionally, the proposed methods (e.g., integrating safety policies, fine-tuning on STAR-1) primarily test existing techniques rather than presenting a novel or effective solution.
>
> The reviewer suggests our work is merely an incremental extension of "reasoning-chain quality assessment" to safety. We respectfully argue that this view overlooks the fundamental difference between **Capability** and **Alignment Integrity**.
> * **Fundamental Shift in Safety Paradigm:** Existing reasoning evaluations focus on logical *correctness* (e.g., math, code). Our work identifies a novel alignment pathology: **Superficial Safety Alignment (SSA)**. This is not simply "wrong reasoning"; it is a specific dissociation where a model internally *fails* to detect risk yet externally *feigns* compliance.
> * **The "Illusion of Safety":** We reveal that high response-level safety (Safe@1 > 90%) often masks low reasoning integrity (Think@1 < 40%). This is not an incremental metric improvement but a **diagnosis of a latent failure mode** that standard evaluations (rewarding safe outputs regardless of the *reason*) actively encourage.
> * **Methodological Challenge:** Constructing BSA was not trivial. It required a multi-stage pipeline with expert verification to ensure reasoning traces contain unambiguous *risk rationales*, distinct from standard QA datasets.
>
> **2. On Related Work Coverage**
> > The related work section should include prior studies that evaluate reasoning-chain quality in LRMs, since the core of this paper lies precisely in measuring reasoning quality from a safety perspective (e.g., Think@k metric).
>
> We agree to include more literature on general reasoning-chain quality. However, we emphasize a critical distinction in the revision:
> * **Distinction:** Prior work evaluates *capability* (can the model reason?). Our work evaluates *trustworthiness* (does the model's reasoning honestly reflect safety principles?).
> * **Gap Filling:** As noted in the paper, current benchmarks are "structurally incapable of detecting internal reasoning failures" in safety. We will cite the suggested works to highlight exactly where they stop and where our safety-specific framework begins.
>
> **3. On Terminological Confusion ("Superficial Safety Alignment")**
> > The term Superficial Safety Alignment overlaps with terminology already used in earlier alignment research, which may lead to confusion. Since the concept here is specifically tailored to reasoning-capable models, a more distinctive and precise naming would improve clarity.
> The reviewer notes a potential overlap with general alignment terms.
> * **Precision of Definition:** We defined SSA specifically for LRMs: "a phenomenon where models produce superficially safe outputs while internal reasoning processes fail to genuinely detect... underlying risks".
> * **Why "Superficial":** The term accurately describes the mechanism: the safety is skin-deep (output-only).
> * **Differentiation:** We explicitly distinguish SSA from *Shallow Safety Alignment* (safe prefix + unsafe body) and *Deceptive Alignment* (intent-explicit) in Section 1 and Appendix C. We will clarify these boundaries further in the introduction to prevent confusion.

---

> ### Author Response · Authors · 2025-12-03
> **Part 2: Response to Reviewer hYdd**
>
> **4. On Novelty of Risk Taxonomy**
> > While the risk taxonomy in the BSA benchmark is well structured, most categories—except Cognitive Shortcut—are not new. This diminishes the overall novelty of the proposed benchmark and the distinctiveness of the problem formulation.
> The reviewer critiques the use of standard risk categories.
> * **Intentional Standardization:** We deliberately aligned our taxonomy with industry standards (e.g., OpenAI, Meta policies) and established benchmarks like BeaverTails/PKU-SafeRLHF. Creating "novel" risk categories (e.g., inventing new types of crimes) would make the benchmark incompatible with real-world safety policies.
> * **The Real Novelty is the *Mechanism*, not the *Label*:** The contribution is not the label "Hate Speech," but the **SSA Scenarios** (Cognitive Shortcut, Risk Omission, Over Sensitivity). These scenarios are engineered to *trap* the reasoning process—something standard taxonomies do not do. The reviewer acknowledged "Cognitive Shortcut" is new; we argue that systematically mapping standard risks to these *new reasoning failure modes* is a significant contribution.
>
> **5. On Lack of Cross-Benchmark Validation (e.g., WildJailbreak)**
> > Experiments are conducted only on the BSA benchmark. The authors should have evaluated models on existing safety benchmarks (e.g., StrongReject, WildJailbreak) as well. If the same SSA phenomena are well observed on standard benchmarks, the necessity and distinct contribution of BSA would be questionable.
> The reviewer asks why we did not evaluate on benchmarks like StrongReject or WildJailbreak. This is a critical point that demonstrates the **necessity** of our work.
> * **Incompatibility of Existing Benchmarks:** Benchmarks like WildJailbreak are **Output-Centric**. They provide ground truth only for the *final answer* (Safe/Unsafe). They **lack the reasoning-layer annotations** required to calculate `Think@k`.
> * **Impossible to Measure SSA:** To diagnose SSA, one *must* compare the correctness of the thought process against the safety of the output ($R_{model} \neq R_{true} \land S(A)=true$). Since WildJailbreak does not annotate the "correctness of the thought process," it **cannot** be used to measure SSA.
> * **BSA's Unique Value:** This limitation of existing benchmarks is precisely *why* we built BSA. BSA is the first benchmark to provide the *Risk Rationale* annotations  necessary to decouple reasoning quality from output safety.
>
> **6. On Formatting Issues**
> > The reference citation format deviates from the conventional ICLR style and should be corrected for consistency.
> We thank the reviewer for pointing this out. We will strictly correct all citation formats to comply with the ICLR style guide in the final version.

---

### Official Review · Reviewer_zRTE · 2025-11-01

**Soundness:** 4
**Presentation:** 2
**Contribution:** 4
**Rating:** 6
**Confidence:** 4

**Summary:**

This paper investigates an important phenomenon: models produce superficially safe outputs while internal reasoning processes harmful content, and summary this as SSA (Superficial Safety Alignment (SSA)). The authors introduce a novel benchmark: Beyond Safe Answers (BSA) with over 2000 instances and report results on 23 reasoning models. The evaluation results show significant emergency to improve reasoning models' internal safety as their think process is also open to the public (except some close-source reasoning models like o3).

**Strengths:**

1. The investigated phenomenon is very important. While some reasoning models do not open-source their thinking process (like o1), others (like r1) expose the full thinking content to the user. This makes the Superficial Safety Alignment (SSA) much serious.

2. The authors systematically summarizes this phenomenon and provides a complete framework to evaluate it, providing a benchmark for future improvements to this issue.

3. The evaluation metric fully considers the cost of human resource evaluation and reports (1) Safe@1  and  Think@1 And (2) Safe@k  and  Think@k In addition, the fairness of LLM as a judge was also reported. This makes the evaluation results convincing.

4. The authors provide a simple method to deal with this issue that finetunes the reasoning models with high-quality data.

**Weaknesses:**

1. The presentation could be improved. Line 127-132 has some space that making this page appear somewhat empty.

2. How many GPU hours, total tokens, and dollar cost does one evaluation pipeline consume?

**Questions:**

Please see the weakness.

---

> ### Author Response · Authors · 2025-12-03
> **Response to Reviewer zRTE**
>
> We thank the reviewer for the thoughtful comments and for recognizing the importance and contribution of our work on Superficial Safety Alignment (SSA) and the BSA benchmark.
>
> ---
>
> **1. Presentation and layout around lines 127–132**
>
> > *"The presentation could be improved. Line 127–132 has some space that making this page appear somewhat empty."*
>
> We appreciate this observation. The visually sparse region around lines 127–132 is not intentional; it is caused by LaTeX's float placement policy near a section boundary, where a figure is pushed to the top of the next page, leaving extra white space in that part of the column.
>
> In the revised version, we will (i) explicitly adjust the placement of the relevant figure and the surrounding section break, and (ii) slightly tighten the local vertical spacing, so that the introduction of BSA and the contribution bullets form a visually coherent block without an "empty" area. We will also do a careful pass over all figures and section breaks to avoid similar layout artifacts elsewhere. These changes only affect typesetting and do not alter any technical content.
>
> ---
>
> **2. Compute footprint: GPU hours, tokens, and monetary cost**
>
> > *"How many GPU hours, total tokens, and dollar cost does one evaluation pipeline consume?"*
>
> We are happy to clarify the resource footprint for the experiments reported in the paper. Our setup separates (i) the **main BSA evaluation pipeline**, which runs entirely via cloud APIs, from (ii) **fine-tuning ablations**, where we train models of different sizes on our safety-oriented data.
>
> **GPU hours.**
>
> All 23 LRMs evaluated on BSA, as well as the LLM-as-a-judge that labels both answers and internal reasoning, are accessed through provider APIs. Thus, the **core evaluation pipeline itself does not consume local GPU hours** on our side; all inference for benchmarking is executed on the providers' infrastructure.
>
> Local GPUs are used **only** for the fine-tuning ablations, where we train several model sizes (small / base / larger variants). Since training time naturally scales with model size, the GPU hours are different for each variant. In total, these fine-tuning experiments require on the order of **ten A100-equivalent GPU hours**, which we will report more explicitly in the appendix.
>
> **Tokens.**
>
> Because we rely on multiple commercial providers and model families, the *global* token count across all models is difficult to reconstruct exactly. However, we can report a reliable lower bound from the logging of our LLM-as-a-judge:
>
> * From the judge logs **alone**, the total number of **input tokens** processed over a full BSA sweep across all 23 LRMs is **on the order of** ≈ one billion.
> * The LRMs' own generations (prompts, chains-of-thought, and final answers), also via API, contribute a comparable additional token volume, but we do not have a precise aggregate count across all providers.
>
> We will add these order-of-magnitude figures (in particular, the ∼10^9 judge input tokens) to the appendix for transparency.
>
> **Monetary cost.**
>
> Using providers' billing records, the **total API expenditure** for the experiments reported in the paper—i.e., one complete execution of the evaluation pipeline over all 23 LRMs on BSA, plus the associated judge calls and repeated runs used in our analysis—is approximately **USD 13,000** in total:
>
> * Roughly **USD 10,000** is spent on **model inference** for the 23 LRMs (generating chains-of-thought and final answers).
> * Roughly **USD 3,000** is spent on the **LLM-as-a-judge**, dominated by the ≈10^9 input tokens mentioned above.
>
> Among the 23 LRMs, some families (in particular **Claude** and **Gemini**) are significantly more expensive per token than others and therefore account for a disproportionate share of the total inference cost.
>
> We will include this breakdown (GPU usage limited to fine-tuning, ∼10^9 judge tokens, and the ≈USD 13k total API cost with ≈USD 10k for model inference and ≈USD 3k for judging) in an additional appendix section, so that future work can clearly see the resource requirements for reproducing or extending our evaluation.

---

### Author Response · Authors · 2025-12-03
**Summary of Rebuttal Updates & Response to the AC**

Dear Area Chair,

Thank you for handling our submission 13570. We appreciate the reviewers' thoughtful comments. This note briefly (1) explains why **Superficial Safety Alignment (SSA)** is a serious safety risk, (2) highlights our core contributions, and (3) summarizes how our rebuttal and revision address the main concerns.

### 1. Why SSA Is Dangerous

We formalize **Superficial Safety Alignment (SSA)**: cases where a model's *answer* looks safe, but its *reasoning* fails to recognize or analyze key risks. This creates a dangerous **illusion of safety**:

- Models can pass output-based safety checks and appear reliable.
- Yet subtle risks (e.g., self-harm cues, secondary weapons implications, data exfiltration paths) may be missed, so small prompt changes, different sampling, or downstream tool use can yield unsafe behavior.

As more **Large Reasoning Models (LRMs)** expose their chain-of-thought, unsafe reasoning is no longer an internal detail but something users see and may act on. For such models, **evaluating only final answers is insufficient**; safety must also be assessed at the reasoning level.

### 2. Core Contributions

**(1) Conceptual: SSA as a distinct alignment failure mode.**
We distinguish SSA from capability errors (hallucination), over-refusal/template-style safety, and deceptive alignment. SSA targets the **alignment integrity of reasoning**: whether the model genuinely identifies and reasons about relevant risks, even when the surface answer is safe. This offers the community a concrete, measurable safety notion beyond “does the answer contain unsafe content?”.

**(2) BSA benchmark: risk-rationale dataset designed for SSA.**
We introduce **Beyond Safe Answers (BSA)**, a 2,000-instance benchmark with **expert-written risk rationales**, constructed to trigger three failure modes—**Cognitive Shortcut**, **Risk Omission**, **Over Sensitivity**—across nine risk categories aligned with major safety policies. Many items are explicitly designed so that a safe-looking answer can be produced **without** full risk reasoning, while correct reasoning must identify **all** relevant risks. Existing safety benchmarks, which only label the final answer, cannot detect such SSA cases.

**(3) Large-scale empirical evidence and mitigation trade-offs.**
We evaluate **23 LRMs** and find SSA is **widespread**: many models achieve **>90% Safe@1**, yet only **~30–55% Think@1**, often lower for Think@k. This reveals a systematic gap between safe-looking outputs and risk-aware reasoning. We then study two mitigation families:

- **Inference-time safety rules**, which reduce Risk Omission but increase Over Sensitivity, quantifying a safety–utility trade-off.
- **Fine-tuning with safety reasoning data**, which improves both response safety and reasoning quality (especially for smaller models), but exposes a **“safety alignment tax”** when pushing for stronger coverage.

BSA and our analyses thus provide **actionable infrastructure** for measuring and improving reasoning-level safety.

### 3. Key Rebuttal Clarifications and Revisions

In response to reviewer feedback, we made the following clarifications and edits:

1. **Novelty and positioning.**
   We strengthened the introduction and related work to emphasize that our main contribution is the **formalization and measurement of SSA**, not generic chain-of-thought evaluation. We explicitly contrast SSA with deceptive alignment, over-refusal, and standard reasoning benchmarks.

2. **Risk taxonomy and SSA scenarios.**
   We clarified that our top-level risk categories are aligned with established safety policies, while the novelty lies in **risk-rationale annotations** and the three **SSA scenarios** (Cognitive Shortcut, Risk Omission, Over Sensitivity) specifically engineered to probe risk-aware reasoning. We added clear explanations and qualitative examples in the main text and appendix.

3. **Cost, practicality, and presentation.**
   We now report approximate token counts and API cost for the evaluation, highlight the release of dataset, prompts, and code. We also refined terminology (reducing anthropomorphic or overly “alarmist” wording), clarified key definitions, and fixed formatting and citation issues.

### 4. Closing

Reviewers broadly agree that the **problem is important and timely**, the **evaluation pipeline is sound and validated**, and the **BSA benchmark is likely to be a useful community resource**. The remaining concerns mainly involve framing and presentation rather than methodological flaws, and our rebuttal and revisions directly address them.

Given the increasing deployment of reasoning-capable models and the real risk of **"safe-looking but unsafely reasoned"** outputs, we believe this work offers timely and practically relevant safety infrastructure for the community. We respectfully hope you will consider it a strong candidate for acceptance.

Sincerely,
The Authors

---

### Meta-Review · Area_Chair_7sqX · 2026-01-12

**Summary:**

This paper introduces Superficial Safety Alignment (SSA), a failure mode in which Large Reasoning Models (LRMs) produce superficially safe answers while failing to correctly identify or reason about underlying risks. To diagnose this phenomenon, the authors propose Beyond Safe Answers (BSA), a benchmark of 2,000 carefully annotated instances spanning three SSA scenarios (Cognitive Shortcut, Risk Omission, Over Sensitivity) and nine safety categories, together with reasoning-level metrics (Think@k, F-Score). An extensive evaluation across 23 LRMs demonstrates a large gap between response-level safety and reasoning-level risk awareness, and the paper further studies mitigation strategies via prompting and fine-tuning.

Reviewers’ concerns primarily centered on borderline novelty and positioning: several felt that SSA/BSA may read as an incremental extension of prior work on reasoning-chain evaluation and safety–capability trade-offs, and were not fully convinced the SSA framing is sufficiently distinct. Reviewers also questioned the scope/generalization of the benchmark (focused on a specific subset of safety failures and evaluated only on BSA), leaving uncertainty about how much additional signal it provides beyond existing safety evaluations. Finally, multiple reviewers flagged presentation/clarity issues (heavy terminology, potential anthropomorphism, and initially unclear distinctions among SSA scenarios). This work would benefit from a significant revision to sharpen contributions and positioning.

**Reviewer Concerns:**

Several reviewers questioned whether SSA constitutes a sufficiently distinct conceptual advance, noting overlap with prior work on reasoning-chain quality, safety reasoning, and alignment tax. While the authors clarified that SSA targets alignment integrity rather than capability, reviewers remained divided on whether this distinction is sharp enough to warrant a new benchmark framing rather than an incremental extension of existing safety evaluation paradigms.

A second concern relates to the scope and generality of the benchmark. BSA focuses on a specific subset of safety failures—cases where reasoning is unsafe but outputs appear safe—and some reviewers felt this captures only one slice of the broader LRM safety landscape. Relatedly, experiments are conducted exclusively on BSA; although the authors argue existing benchmarks cannot measure SSA by construction, reviewers remained uncertain about how much new signal BSA provides beyond known safety–capability trade-offs.

Finally, there were concerns about presentation and clarity: heavy terminology, potential anthropomorphism, and initially insufficiently clear distinctions between SSA scenarios (CS vs. RO vs. OS). The rebuttal addressed many of these issues with clearer definitions, added explanations, cost disclosures, and improved positioning, but these clarifications do not fully eliminate concerns about contribution depth relative to venue standards.

Overall, while the work is careful, well motivated, and likely useful to the community, reviewers were split on whether the contribution clears the bar for acceptance at this venue rather than being better positioned as a refined benchmark or follow-up study.

**Reviewer Scores:**

- zRTE: Likely no change (already marginally positive but explicitly borderline).
- hYdd: Likely no change; concerns about novelty and scope persist despite clarifications.
- upTN: No change.
- hcqa: No change.

---

### Decision · Program_Chairs · 2026-01-26

Reject